# Effects of 3D Moving Platform Exercise on Physiological Parameters and Pain in Patients with Chronic Low Back Pain

**DOI:** 10.3390/medicina56070351

**Published:** 2020-07-15

**Authors:** Soochul Kim, Yongseok Jee

**Affiliations:** 1Department of Education Major of Physical Education, Graduate School of Education of Hanseo University, Hanseo 1-Ro, Haemi-myeon, Seosan 31962, Korea; kkpo92@naver.com; 2Department of Leisure and Marine Sports, Hanseo University, Hanseo 1-Ro, Haemi-myeon, Seosan 31962, Korea

**Keywords:** 3D moving platform, pain, static muscle contraction, dynamic muscle contraction

## Abstract

*Background and objectives:* Patient-handling activities predispose women to chronic low back pain (CLBP), but sufficient evidence is not available on whether a 3D moving platform, made for core stability exercise, affects pain, trunk flexibility, and static/dynamic muscle contractions in CLBP patients. *Materials and Methods:* The participants were twenty-nine women who were randomly divided into a control group (CON) and a 3D exercise group (3DEG), which took part in 3D moving exercise three times a week for 8 weeks. Both groups measured a visual analog scale (VAS) about their CLBP. Body composition, forward and backward trunk flexibilities, static muscle contraction property in rectus abdominis, and erector spinae were measured by tensiomyography, which found contraction time (Tc) and maximal displacement (Dm). Dynamic muscle contraction property in the abdomen and back were measured with an isokinetic device, which could measure peak torque (Pt) and work per repetition (Wr), before and after the trial. *Results:* The 3DEG had a significantly decreased fat mass and waist/hip ratio, as well as improved static muscle contractions of the erector spinae. The Wr of trunk extensor of 3D exercise group were also significantly increased. In the VAS, although the scores showed a significant change in some variables, while others did not. The Δ% in feeling pain at rest or at night, during exercise, walking, sitting in a hard chair, sitting in a soft chair, and lying down in 3DEG were significantly changed after 8 weeks. This indicates that the platform exercise provided a greater reduction of pain for activities that are done on a daily basis. *Conclusions*: This study confirms that the 3D moving platform exercise can provide the similar effect of the core stability exercise used in previous studies. Moreover, this study suggests that 3D moving platform exercise is a suitable means to reduce fatness, to increase trunk extensor, and to increase trunk backward flexibility, which led to reduced back pain in the women with CLBP.

## 1. Introduction

Most individuals experience low back pain (LBP) in their lives. For approximately one-half of them, the pain is self-limiting, but in about 10–50% of patients, LBP lasts more than 12 weeks, which is considered chronic low back pain (CLBP) [1,2]. Among many factors causing low back pain, a lack of physical activity has been viewed as a primary risk factor, which results in weakness of the lower back muscles and the abdominal muscles that connect the lumbar spine. There are an increasing number of patients who complain of pain in the musculoskeletal system around the spine due to poor habits, such as improper posture and a prolonged sedentary lifestyle. In other words, the number of people who suffer from spine disorders is increasing rapidly every year. Such lumbar-related disorders are more prominent in women than in men, with the weakening of the musculoskeletal system making women prone to this disease, when they do not engage in physical activity [3,4,5,6,7].

Spinal problems caused by poor posture lead to persistent fatigue and pain, causing many obstacles to the individual’s daily life, and increasing medical and social costs. Thus, CLBP care requires deeper understanding [8]. Several studies have reported that problems due to bad posture and pain, due to musculoskeletal disorders associated with the spine, contribute to the deformation of the spinal alignment [9,10]. Spinal malalignment is a deformity of the normal spinal alignment, and types of spinal malalignment include lordosis, kyphosis, scoliosis, forward head posture, and pelvic torsion. The alignment of the pelvis and lumbar vertebrae affects other parts of the body, which is likely to cause further health problems.

Core stability exercises (CSEs) that improve lumbopelvic stability may be included as a part of prevention and clinical rehabilitation for patients with CLBP. CSEs include a range of exercise programs with different approaches, all of which have the common goal of improving lumbopelvic and abdominal control. These exercises are designed to enhance the ability of the neuromuscular and motor control systems to prevent spinal injury [11].

The recently developed 3D moving platform is also a kind of CSE, but its effectiveness has not been proven. The 360-degree rotational motion function of the 3D moving platform is designed to fit the natural spiral motion of the body and strengthen the muscles around the body by transmitting the exercise power to the deep muscles that the existing linear reciprocating motion does not reach. This ensures stability by establishing various three-dimensional railings, and can maximize pain relief, as well as corrective treatment through muscle extension and contraction. It is a device that helps to properly align the deformed joint and the twisted spine by applying a sling or harness assistive device. The 3D moving platform has the advantage of providing the ability to do exercises that can stimulate deep muscles of the lumbar on the machine itself, without the care of the therapist. According to recent studies, CSEs are effective in short-term pain reduction and physical function improvement in patients with CLBP [12]. Several studies revealed that CSEs increases the cross-sectional area of the lumbar multifidus in women with CLBP and significantly increases the thickness of transversus abdominis in patients with CLBP during rest and muscular contraction [13,14]. In fact, CSEs programs can be helpful for treating women with CLBP. As such, CSE can improve various properties related to the lumbar joint, and furthermore, it is thought that the 3D platform developed for CSE can also perform the functions of CSE. However, until now, it is not known whether the use of this device has a similar effect to CSE. In other words, to the best of our knowledge, no study has revealed whether such a systematic exercise program can affect the pain, trunk flexibility, and static/dynamic muscle contractions of the muscles involved in lumbopelvic stability. Therefore, this study used a single-blinded randomized controlled trial to study the CSE-effects of a 3D moving platform exercise program in women with CLBP.

## 2. Materials and Methods

### 2.1. Study Design and Participants

This study was conducted in a rehabilitation center of Hanseo University in the Republic of Korea. The participants were all women and their age ranged from 20 to 23. Their mean age was 21.06 ± 0.44 years old. The goal of this study was to recruit participants who had CLBP and had not exercise regularly for over six months. This study also included women without prior operation and dysfunction in the spine, and without musculoskeletal disorders or cardiovascular problems. The participants were excluded if they had received any treatment or medication known to affect physical conditions, or if they had undergone any major surgery during the year before the start of this study. The following were also reasons for exclusion: having a history of coronary arterial disease or cerebrovascular disease, an impairment of a primary organ system, severe lung disease, severe cerebral trauma, uncontrolled hypertension, cancer, or a psychiatric disorder. Prior to the study, the participants received detailed explanations regarding all of the procedures in this study, and were then asked to complete questionnaires, which included basic demographic questions and a visual analogue scale (VAS) for CLBP.

Thirty-two participants were initially screened to determine eligibility for the study. However, one participant was excluded because of personal reasons, one participant dropped out in the allocated assessment stage, and two people did not complete the follow-up stages. Finally, twenty-nine participants were enrolled in this study. After taking baseline measurements, participants were randomly assigned to one of two groups: the 3D platform exercise group (3DEG, *n* = 14) and the control group (CON, *n* = 15), as shown in Figure 1. Complete subject characteristics of this study are presented in Table 1.

### 2.2. Research Ethics

This study was conducted in accordance with the Declaration of Helsinki, and was approved by the ethics committee (6 February 2020 to 6 February 2021: 2-7001793-AB-N-012019119HR). Written informed consent was obtained before enrollment. All of the participants signed an informed consent form and completed a self-report questionnaire, including a VAS, which is a tool for measuring the degree of pain felt in the lower back from a comfortable position to an active position [15]. Then, they were assessed using a physical examination to measure body composition and trunk flexibilities. Tensiomyography (TMG) and isokinetic strength tests were also administered.

### 2.3. Body Composition Measurements

A BMS 330 anthropometer (Biospace Co., Ltd., Seoul, Korea) was used to measure height. An Inbody 230 (Biospace Co., Ltd., Seoul, Korea) analyzer was used for the body composition measurements with the bioelectrical impedance analysis method. The participants were asked to remove all metal objects and anything else that might interfere with the electric stimuli, including socks, before stepping on the platform. They were also asked to hold onto the handles and stand still for 3 min. The participants abstained from food, exercise, and diuretic drinks for 4 h, 12 h, and 7 days, respectively, prior to assessments. The participants were also asked to void 30 min prior to the assessment [16].

### 2.4. Forward and backward Flexibilities of Lumbosacral Joint Measurements

Lumbar flexibility was assessed using trunk forward flexion and backward extension tests [17]. Before assessment, each subject performed a standardized warm-up consisting of 5 min of stretching exercises. The extent of trunk flexion and extension was measured by a flexibility meter, model TKK1859 (Takei Inc., Tokyo, Japan). Trunk flexion was assessed using the sit and reach test. Participants performed the test with the legs fully extended and knees relaxed. They were required to extend their arms as far as possible and hold at the furthest point for 2 s. After completion of the trunk flexion test, all participants were assessed in the trunk extension test. The participants were in a prone position with their hands clasped behind their head. Another examiner stabilized their ankles, instructed them to raise their back upward, and then recorded the point on the tester. The participants were required to hold their position at the topmost point for 2 s.

### 2.5. Static Muscle Contraction Measurements

A TMG device (TMG-S1, TMG-BMC Ltd., Ljubljana, Slovenia) is a static muscle contraction measure, based on the quantification of radial muscle displacement in response to a single electrical stimulus. We measured the rectus abdominis and erector spinae for trunk musculature. The extraction of contractile parameters from TMG responses is straightforward, and does not require special post-processing or filtering [18]. Moreover, this TMG measurement has been suggested to be only slightly affected by longitudinal tendon and ligament elasticity, serial connective tissue, joint friction, and extremity inertia [19]. TMG measurements are generally performed in a static and relaxed position, which needs a digital transducer. Electrical stimulation is delivered with two surface electrodes placed proximally and distally to the sensor tip. TMG assesses muscle mechanical responses, based on radial muscle belly displacement induced by a single electrical stimulus between the proximal and distal parts of the rectus abdominis and erector spinae. In the case of the rectus abdominis, the sensor tip was placed at a point 3 cm away from the left and right sides of the navel. Electrodes were attached 3 cm apart, proximal and distal from the sensor tip, which served as the center point. The degree of muscle contraction was measured at the radial muscle belly. The position of the sensor tip of the erector spinae was determined to be 5 cm above the lumbosacral joint. It was positioned 3 cm away from the left and right sides of that point, and the electrodes were attached 3 cm apart, proximal and distal from the sensor tip. The electric stimulation provided under increasing electrical current intensities was between 10 mA to 65 mA, and the length of the stimulation was one millisecond (ms). An isometric contraction was generated by the electrical stimulation. Electric stimulation was given in 10 mA increments until maximal displacement was reached. For reference, maximal displacement (Dm) decreases when there is damage to the muscle, while Dm increases when muscle damage recovers. Displacement-time curve recordings allow muscle contractile properties to be assessed, which include: the following delay time, contraction time (Tc), sustain time, and relaxation time [18]. From these four parameters, Dm and Tc are generally considered the most valid factors [19,20]. The normal curve from TMG has a steep shape and Tc appears at short intervals. Characteristically, the shape of the overall curve appears to collapse after an injury, which is due to the fact that the muscle contraction does not proceed normally and rapidly, as shown in Figure 2.

### 2.6. Dynamic Muscle Contraction Measurements

Participants were positioned in a standing position on the isokinetic dynamometer (HUMAC^®^/NORM^TM^ Testing and Rehabilitation System, CSMi, Stoughton, MA, USA), according to the guidelines for evaluating trunk extension/flexion (TEF) [21]. All participants stood on the footplate of the TEF modular component, their heels were placed against the footplate heel cups. The height of the footage had kept on coordinating until the length of the rubber alignment indicated 3.5 cm below the top of the iliac crest for alignment of their vertical axis with a dynamometer to adjust. The pelvic belt was loosely fastened across the top of the anterior superior iliac spines. The popliteal pad height was adjusted to a position directly behind the patellae at the popliteal space. The lower body was stabilized using the tibial, popliteal, and thigh pads with the knees slightly bent. The participants leaned against the sacral pad and were moved forward or back via the fore-after alignment wheel, until the rubber alignment pointer was centered approximately at the axis of rotation. For the TEF test, the upper leg and the lower leg should be fixed to prevent anterior protrusion of the lower limb. The fixed position of the upper leg was such that the bottom of the upper leg pad was aligned with the top of the patella, and, in the case of lower leg, the top of the lower leg pad was aligned with the bottom of the patella. Once those pads were aligned, the locking lever was secured. In order to fix the upper body, a pad was wrapped around the chest, while the subject held the handles for additional stability. At this time, the lower surface of the upper body pad was measured, and then fixed to coincide with the inferior angle of the scapula. In other words, the chest pad was placed in a position that was parallel to the scapular pad and secured. The range of motion (RoM) of TEF was approximately −15° to 95°. The participants performed 4 maximal warm-up repetitions and 5 maximal test repetitions at 30°/s, which gained peak torque (Pt). They then performed 4 maximal warm-up repetitions and 15 maximal test repetitions at 90°/s, which gained work per repetition (Wr). The rest time between tests was 60 s. All tests were supervised by only one trained researcher.

### 2.7. Back Pain Measurements

A VAS was used to measure pain variables related to body movement, that is, pain in the night, pain in exercise and walking, pain in lying or standing, and stiffness in the waist were selected and measured. All of the participants were asked to rate how they felt pain in their back using a bipolar rating scale, which is a bar-shaped box with a height of 5 cm and a length of 10 cm. The pain scale ranged from no pain (close to “0”) to severe pain (close to “10”). After the participants marked within the box, a transparent paper with score indicator was placed on top of the boxes to obtain a numerical score. The participants were evaluated by a professional psychologist at the beginning and end of the study [15,22]. The reliability of the questionnaire was measured by calculating Cronbach’s α, representing internal consistency. The Cronbach’s α of the pain scale was 0.921.

### 2.8. Rehabilitation Program through 3D Platform Intervention Measurements

All participants agreed not to change their daily activity patterns outside of their participation in this study. Participants also agreed not to change their dietary habits throughout the study period. The 3DEG took part in a supervised progressive rehabilitation program for 3 days (Monday, Wednesday, Friday) a week for 8 weeks, as shown in Table 2. They participated in the rehabilitation program while the platform was operating, while CON participated in the rehabilitation program when the platform was not operating. The rehabilitation program on the 3D platform consisted of various types of exercises for stretching the core muscles and for strengthening the paraspinal muscles. First, both groups began to warm-up conditioning with upper/lower extremity (5 min) by a therapist for 8 weeks. Second, they performed three work-out sessions with a 3D moving platform (MS-3000, Medical Science Co. Ltd., Seoul, Korea). The dimensions and weight of this 3D platform are 1180 × 1900 × 2360 mm and 210 kg. The speed, angle including control program, and rating voltage (power) of 3D platform are 0–34 rpm, 1–16 step, and 60 Hz (750 watt), respectively.

This was followed by the 1st work-out phase (Day 1 to Week 2), which involved stabilizing the trunk, strengthening the hip and hamstrings, and stretching the piriformis and gluteus under ratings of perceived exertion (RPE) 13 (somewhat hard). The goal of this stage focused on softening the body, tolerating weight bearing, improving RoM, and reducing pain. Next, all of the participants took part in the 2nd work-out phase (Week 3 to Week 5), a 3D platform intervention, which focused on strengthening hip adductors, improving trunk stabilization and coordination, and increasing core muscle and motor control. The goal of this stage focused on tolerating full weight bearing and improving passive RoM and neuromuscular control. The last phase was the 3rd work-out phase (Week 6 to Week 8), during which, the exercises from the 2nd work-out phase increased the intensity (RPE 13–15) and duration (over 40 sec/set). Moreover, the motions of the 3rd work-out phase were applied with three types of standing positions. The goal of this stage focused on improving spinal anterior-, posterior-, and lateral-chain activations, and on maintaining balance ability and proprioception. The rehabilitation exercises and their repetitions and sets used in this study were extracted from results from previous researches [23,24,25].

### 2.9. Data Analysis

The results obtained through the experiment were input into Microsoft Excel (Microsoft, Redmond, WA, USA), and calculated using technical statistics (mean ± standard deviation). The SPSS program (version 18.0; SPSS Inc., Chicago, IL, USA) was used to calculate statistics for this study. The distribution of all data was checked using the Shapiro–Wilk test. Prior to analysis, we observed the difference between groups through Mann–Whitney U test before comparing between groups as shown in Table 1. An analysis of variance (ANOVA) test was used to evaluate the significance of the differences between groups at baseline. Then, the effects of the interventions were assessed using an analysis of variance for repeated (2 × 2) measures (group, time, and group by time interaction). An intention-to-treat analysis was performed to compare the intervention group (3DEG) with the CON. The between-group factor was the study groups (i.e., 3DEG vs. CON) and the within-group factor was the week (i.e., Week 0 vs. Week 8). When the variables between times were further analyzed, the Δ% was calculated. The level of statistical significance chosen was *p* ≤ 0.05.

## 3. Results

### 3.1. Difference in Anthropometric Indices

There were no differences in baseline characteristics between 3DEG and CON, which indicates homogeneity was established. As shown in Table 3, the participants showed similar results, except for fat mass and waist/hip ratio (WHR). In detail, although the Δ% of fat mass in CON increased ≈ 2.3%, that of 3DEG decreased ≈ 4% (not shown in the table). This indicates that the platform exercise provided a greater amount of exercise that can further reduce fat. These results indicate that there was a significant difference in time (*p* < 0.05). Similar to the results of fat mass, the WHR of CON did not change significantly after 8 weeks, whereas that of 3DEG decreased ≈ 1.25% (not shown in the table), which was significantly different in time (*p* < 0.01) and group by time (*p* < 0.05).

### 3.2. Effect of 3D Platform Exercise On Static Muscle Contraction

As shown in Table 4, there were no significant changes in the left or right Tc and Dm of the CON. Similarly, there were no significant changes in the left or right Tc and Dm of the 3DEG after 8 weeks. However, the increased right Tc of the erector spinae in CON was greater than that of the erector spinae in 3DEG. These results indicated that there was a significant difference in time (*p* < 0.05). This indicates that the platform exercise provided a greater amount of exercise that can increase the contraction of muscles and improve balanced development of the left (20.00 ± 9.32 ms) and right (24.5 ± 15.55 ms) erector spinae muscles after 8 weeks.

### 3.3. Effect of 3d Platform Exercise on Dynamic Muscle Contraction

As shown in Table 5, there were no significant changes in most variables of isokinetic moments at 30°/s in both groups after 8 weeks. However, the Wr of trunk extensor at 90°/s in the 3DEG was significantly increased in the trunk extensor, but this was not changed in the CON. More specifically, although the Δ% of the Wr of trunk extensor in CON increased ≈ 2.1%, that of 3DEG increased ≈ 22.5% (not shown in the table). This indicates that the platform exercise provided a greater amount of exercise that can increase dynamic contractions of the trunk extensor. These results show that there was a significant difference in time (*p* < 0.05).

### 3.4. Effect of 3D Platform Exercise on Trunk Flexibilities

As shown in Table 6, although the trunk forward flexibility of 3DEG tended to increase, it tended to decrease in the CON, although there was not a significant difference. However, the trunk backward flexibility in both groups tended to increase. In detail, the Δ% of the backward flexibility in CON increased ≈ 2.8%, while that of 3DEG increased ≈ 9.1% (not shown in the table). This indicates that the platform exercise provided a greater amount of exercise that can further soften the trunk extensor muscles. These results show that there was a significant difference in time (*p* < 0.05).

### 3.5. Effect of 3D Platform Exercise on Visual Analogue Scale

As shown in Table 7, the VAS scores in both groups tended to improve, although the VAS scores showed a significant change in some variables, while others did not. In detail, the Δ% of back pain in CON decreased ≈ 53.8%, while that of 3DEG decreased ≈ 80.5% (not shown in the table). The Δ% in feeling pain at night, during exercise, walking, sitting in a hard chair, sitting in a soft chair, and lying down in CON were changed by ≈ 9.2%, ≈ −31.6%, ≈ −13.1%, ≈ −26.4%, ≈ −21.2%, and ≈ −16.8%, respectively. The Δ% in feeling pain at night, during exercise, walking, sitting in a hard chair, sitting in a soft chair, and lying down in 3DEG were changed by ≈ −48.2%, ≈ −62.1%, ≈ −32.9%, ≈ −45.4%, ≈ −35.3%, and ≈ −41.9%, respectively (not shown in the table). This indicates that the platform exercise provided a greater reduction of pain for activities that are done on a daily basis. There were significant differences in back pain (*p* < 0.05, within time), night pain (*p* < 0.05, between group; *p* < 0.05, within time), exercise (*p* < 0.01, within time), walking discomfort (*p* < 0.05, within time), hard chair (*p* < 0.05, within time), soft chair (*p* < 0.05, within time), and lying down (*p* < 0.05, within time), respectively.

## 4. Discussion

This study revealed that 3D moving platform exercise can be effective in managing the CLBP of women. After 8 weeks of platform exercise, the fat mass and WHR in the 3DEG was improved, although it remained unchanged in CON. Furthermore, the platform exercise improved the erector spinae Tc of the static muscle contraction, extensor Wr of the dynamic muscle contraction, trunk backward flexibility, and some of VAS in the 3DEG, but did not in CON.

Many studies have been published on the application of exercise to treat CLBP. In the aspect of body composition in LBP, Ogden et al. [26] reported regular exercise is vitally important for losing a body weight and is a predominant component in lifestyle modifications for the patients with low back pain. Roffey et al. [27] and Thompson et al. [28] reported that exercise can lead to decreased subcutaneous adipose tissue and reduced visceral adiposity in the LBP patients. This is thought to occur via a regular exercise and similarly to the result of this study. That is, the CSEs via 3D platform exercise can decrease the visceral fatness of abdomen. In addition to changes in body composition, exercise, including CSEs, is important to improve musculatures around the lumbar joint in LBP patients. Hodges [11] has suggested that the main mechanism of exercise therapy for managing CLBP must improve the neuromuscular function and strengthen the muscles that control and support the spine and pelvis. Akuthota et al. [23] suggested that core stability is essential for proper load balance within the spine, pelvis, and kinetic chain. The so-called core is the group of trunk muscles that surround the spine and abdominal viscera. Noormohammadpour et al. [29] also reported that CSEs could be effective in retraining the trunk muscles, which have an important role in the stabilization, coordination, and control of the spine. Similarly, the 3D moving platform used in this study was created by devising CSEs. However, no study has revealed whether it can affect the physical conditions, including RoM and muscular contraction of the abdominal and back muscles involved in lumbopelvic stability. In addition, no study has confirmed whether it can improve the back pain in CLBP patients. According to the results of static muscle contractions of the abdominal and back muscles of this study, the 3D moving platform exercise improved the erector spinae observed by TMG. In detail, although there was not significantly changed in Tc and Dm of left erector spinae in the 3DEG, the Tc of right erector spinae in the same group was significantly increased after an 8-week intervention. More importantly, the Tcs of left and right erector spinae were balanced at the end of experiment. In other words, by exercising on the 3D platform, we could see the balanced effect that the Tc on the right rises and resembles the Tc on the left in erector spinae muscles.

Similarly, in the results of dynamic muscle contractions of the abdominal and back muscles of this study, there were no significant changes in almost variables of isokinetic trunk flexor in 3DEG after 8 weeks. However, the trunk extensor of 3DEG was significantly increased, whereas those of CON were not changed in the post-values. The 3D moving platform exercise program used in this study was more effective for trunk extensor than trunk flexor. Koumantakis et al. [5] reported that the patients with 8 weeks of recurrent LBP improved through trunk muscle stabilization training. Norris and Matthews [30] also reported that patients with CLBP improved after a 6-week integrated back stability program composed of optimizing posture, back fitness, and functional exercises. In the last few years, TMG measurements have been successfully implemented in different muscle groups to investigate muscle atrophy, muscle endurance, and abdominal muscle stiffness [31,32]. Some of the muscular contractile parameters by TMG have been found to correlate with Pt and with distribution of fiber types in human muscles [19,32]. The results of TMG measurements in this study were the first to our knowledge to identify changes in the trunk muscles through exercise intervention in CLBP patients.

This study also found that 3D platform exercise could decrease abdominal fat in women with CLBP. In the anthropometric indices of this study, although the fat mass and WHR of the CON were not changed, it was significantly changed in the 3DEG after 8 weeks. It is thought that 3D platform exercise provides deeper muscle activation on the rotating platform, and provides a higher metabolic energy consumption effect than CON, which was in motion while the platform was stationary. In other words, we think the 3D platform exercise contributes to fatness reduction by providing metabolic effectiveness, in addition to changes in static or dynamic strength. A study reported that some individuals with CLBP exhibit a reduced aerobic capacity compared with healthy individuals [33], but, as with flexibility and strength, cardiovascular performance is strongly influenced by activity-related increases in pain intensity during testing and, therefore, poor performance may not indicate real impairments in cardiovascular function [34]. Regardless of the reason for diminished performance, improving endurance is a reasonable exercise goal for patients with CLBP [35].

In this study, although there was significant difference in the trunk forward flexibility between groups at baseline, there was no significant difference after 8 weeks. These results suggest that 3D moving platform exercise may reduce waist circumference and increase waist flexibility. Kennedy and Noh [36] reported that a comprehensive rehabilitation program could correct a trunk flexibility and strength deficits through subsequent progression to functional exercises. In other words, they supported the use of exercise as a therapeutic tool to improve impairments in back flexibility and strength. In fact, spine rehabilitation programs including exercise are typically designed around the goals of strengthening the low back [37], increasing trunk flexibility [38,39], and improving cardiovascular endurance [33].

The findings of this study are almost consistent with the results of previous studies; the lack of change in body composition was anticipated, due to the short length of the intervention period. In this study, 3D moving platform exercise improved VAS for back pain in patients with CLBP. Specifically, although there was no significant change in the back pain of the CON, that of the 3DEG was significantly decreased after 8 weeks. The pain level felt through the expansion of the narrow nerve path may have been reduced, because the CSE performed on the 3D platform improved spine alignment by applying more muscle stretching and further stimulating the deep muscles. Manniche et al. [40] reported 105 individuals with CLBP were randomly assigned to high or low intensity exercise, or control groups. Exercise groups consisted of isotonic prone back extension exercises on a bench and latissimus pull down exercise; the high-intensity group performed five times as many sets as the low intensity group. The control group performed light floor exercises. Outcomes were assessed using the low back pain rating scale composite score for pain, disability, and physical impairment. The high-intensity exercise group demonstrated significantly greater improvements in the low back pain rating scale, compared with the low intensity exercise and control groups. Moreover, Risch et al. [41] reported that individuals with CLBP were randomized to receive lumbar extensor strengthening exercise and control groups. The lumbar strengthening group performed isolated lumbar extensor progressive resistance exercises on a variable resistance dynamometer machine 1 to 2/week for 10 weeks. The control group was wait listed and received no intervention. Outcomes included pain intensity, psychosocial function, and lumbar extensor strength. At 10 weeks, the lumbar strengthening exercise group displayed significantly greater improvements in pain intensity, lumbar extensor strength, and psychosocial function on one of several scales. The results of previous studies indicate that sufficient exercise in low back pain patients is an effective means of decreasing pain and reducing disability [42,43,44]. Meanwhile, Rainville et al. [39] suggested that exercise is safe for individuals with LBP, because exercise does not increase the risk of future back injuries or work absence. Noormohammadpour et al. [29] also reported that a multi-step core stability exercise program could be a helpful treatment option for improving quality of life and reducing disability and pain in women with CLBP. Koumantakis et al. [5] revealed that 8-week core stability training decreased pain degree. Improved pain scores during lumbar stabilization exercises in patients with CLBP could be suggested as an integrated mechanism for improving blood flow, releasing spasm, and decreasing the inflammation of local tissues in the lumbar spine, which, in turn, reduced pain [23,45]. These findings are consistent with our study results, that is, the 3D moving platform exercise program used in this study can be said to provide the effect of the core stability exercise used in the aforementioned studies. The subjects in this study had no structural deformations, such as a lordotic or kyphotic spine. However, with the observation that the left and right paraspinal muscles become balanced through the 3D platform movement, it is possible that this can provide help to some extent for individuals with structural diseases. Ultimately, it is thought that the static muscle contraction ability and the dynamic muscle contraction ability developed simultaneously through the core exercises on the 3D platform. These results are supported by previous studies [19,32]. In other words, it was observed that the muscles in the back of the lumbar region were improved, whether static or dynamic, and characteristically, the part lacking muscle function on either side of the back was improved. This provides information that can be used clinically when applying the 3D platform to patients with low back pain.

## 5. Conclusions

This study confirms that the 3D moving platform exercise used in this study can provide the similar effect of the core stability exercise used in previous studies. Moreover, this study suggests that 3D moving platform exercise is a suitable means to reduce fatness, to increase the trunk extensor, and to increase trunk backward flexibility, which led to reduced back pain in the women with CLBP. These changes observed in this study included subjects who chronically complained of low back pain due to the narrowing of the neural tube from the deformation of the muscles and ligaments around the lumbosacral joint. However, it was confirmed that the CSE exercise on the 3D platform performed for 8 weeks can effectively activate the paraspinal muscles, by providing greater stimulation and RoM around the lumbosacral joint than normal CSE. However, this study has some limitations, in that the number of subjects were small, consisted entirely of women, and was conducted over a rather short period of 8 weeks. Addressing the three limitations mentioned above may be helpful for future studies.

## Figures and Tables

**Figure 1 medicina-56-00351-f001:**
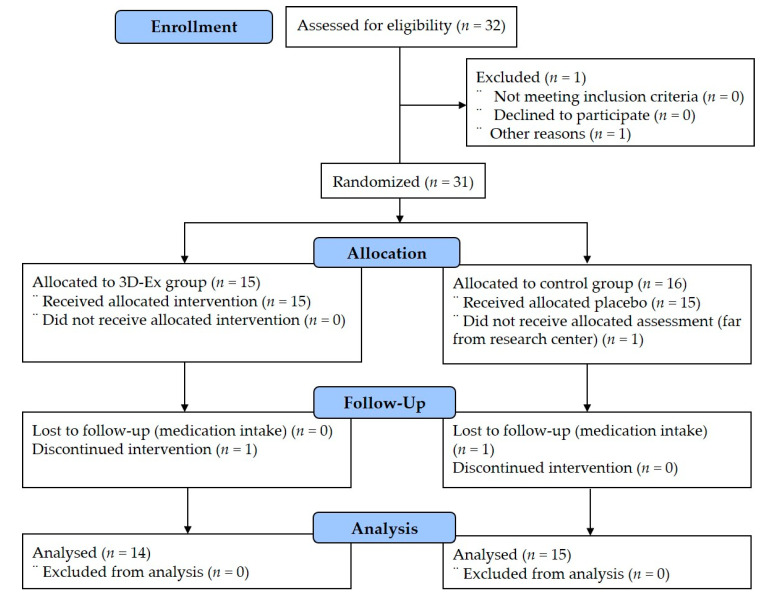
Allocation of participants (consolidated standards for reporting of trials flow diagram).

**Figure 2 medicina-56-00351-f002:**
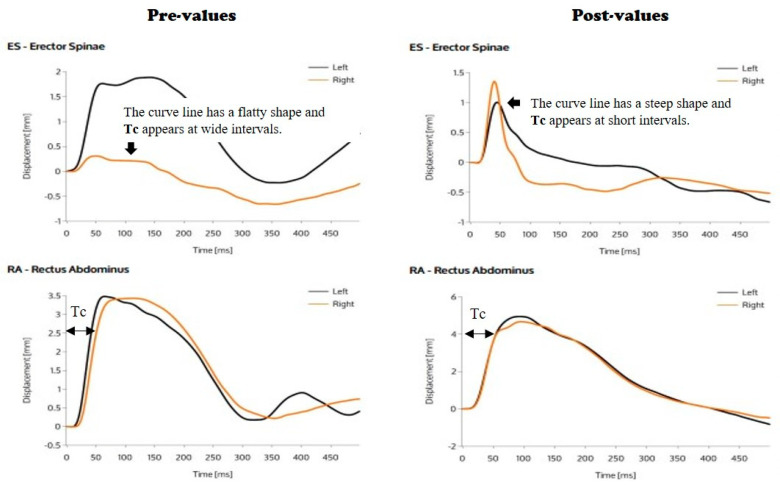
Maximal displacement (Dm) and contraction time (Tc) of erector spinae (ES) and rectus abdominis (RA) in the tensiomyography (TMG). This data is the actual record of Tc and Dm measured in ES and RA of the 3D exercise group (3DEG).

**Table 1 medicina-56-00351-t001:** Physical characteristics of the patients.

Variables (Unit)	Groups		
	3DEG (*n* = 14)	CON (*n* = 15)	Z
Age (y)	20.42 ± 0.67	20.00 ± 0.01	−1.871
Height (cm)	167.83 ± 2.21	168.11 ± 2.98	−0.322
Weight (kg)	54.41 ± 5.85	55.67 ± 4.93	−0.355
LBP history (month)	14.23 ± 1.36	14.52 ± 0.95	−1.425

All data represents the mean ± standard deviation. LBP, 3DEG and CON mean low back pain, 3D platform exercised group and control group, respectively. Z means the value analyzed by Mann–Whitney U test.

**Table 2 medicina-56-00351-t002:** Rehabilitation programs for strengthening and softening the paraspinal muscles.

Types (Periods)	Program Types	Explanation (Intensity/Time)
Warm-up (Day 1 to Week 8)	Stretching in upper and lower extremity on a standing posture	
1st work-out (Day 1 to Week 2)	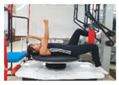	Platform angle/speed for trunk stabilization: 2~4/4~12 Lie in a supine position on a round platform while their knees are bent. Contract lumbar paraspinal muscles while lifting left arm and right leg off the platform. Hold it for 10 s, take rest for 10 s, and repeat 10 times.
	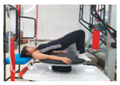	Platform angle/speed for strengthening hip and hamstrings: 2~4/4~12 Lie in a supine position on a round platform and lie on the back with knees bent. Contract lumbar paraspinal muscles while lifting buttocks off the rotating platform. Hold it for 10 s, take rest for 10 s, and repeat 10 times.
	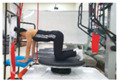	Platform angle/speed for trunk stabilization: 2~4/4~12 Stay on their hands and knees. At this time, the posture is maintained while the platform rotates so that the vertebrae are not bent or stretched. Hold it for 10 s, take rest for 10 s, and repeat 10 times.
	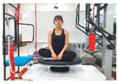	Platform angle/speed for stretching piriformis and gluteus: 3~6/3~6 Sit on the platform while their knees are overlaid, and their back should be straightened. Tilt pelvis from front to back and vice versa while the platform is rotating and contract the lumbar paraspinal muscles. When the platform is tilted backward, stretch their body forward. Prevent the pelvis from tilting and lumbar rotation and angulation. Hold it for 5 min.
2nd work-out (Week 3 to Week 5)	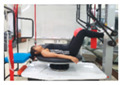	Platform angle/speed for strengthening hip adductors: 3~6/3~6 Lie on their back while putting their legs on the platform. Raise their feet on the railing. Maintain their squeezing a ball by legs while rotating the platform. Hold it for 10 s, take rest for 10 s, and repeat 10 times.
	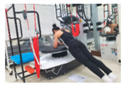	Platform angle/speed for trunk stabilization: 2~4/4~12 Take a plank posture with both arms on the platform. Keep the trunk straight while the platform is rotating. Hold it for 15 s, take rest for 10 s, and repeat 10 times.
	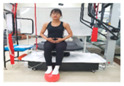	Platform angle/speed for pelvic tilt coordination: 2~6/4~6 Sit on the platform with the two legs straight. Place an air cushion on the plate, and place a ball on the feet to induce an unstable condition. Maintain their postures while the platform is rotating and contracting lumbar paraspinal muscles. Hold it for 5 min.
	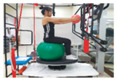	Platform angle/speed for core muscle and motor control: 3~4/3~4 Sit on the Swiss ball in the platform. Try to keep their balance and hold to lumbar paraspinal muscles contraction for 10 s while keeping their back straight. Then lift both of their arms. Hold it for 15 s, take rest for 10 s, and repeat 10 times.
3rd work-out (Week 6 to Week 8)	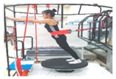	Platform angle/speed for posterior chain activation: 4~6/4~6 Stand on the platform. The sling is put on the abdomen and remains standing straight. Hold it for 40 s, take rest for 15 s, and repeat 10 times.
	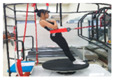	Platform angle/speed for anterior chain activation: 4~6/4~6 Stand on the platform. The sling is put on the back and remains standing straight. Hold it for 40 s, take rest for 15 s, and repeat 10 times.
	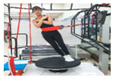	Platform angle/speed for lateral chain activation: 4~6/4~6 Stand on the platform. The sling is put on the lateral pelvis and remains standing straight. Hold it for 40 s, take rest for 15 s, and repeat 10 times.
Cool-down (Day 1 to Week 8)	Stretching in upper and lower extremity on a supine posture	

**Table 3 medicina-56-00351-t003:** Comparative results of body composition.

Items (Units)	Time (T)	Group (G)		ANOVA (*F*)		
		3DEG	CON	G	T	G × T
Weight	Pre	54.41 ± 5.85	55.67 ± 4.93	0.291	1.208	0.008
(kg)	Post	54.13 ± 5.76	55.43 ± 4.66			
Muscle mass	Pre	21.07 ± 1.91	21.32 ± 1.23	0.092	1.799	0.072
(kg)	Post	21.32 ± 1.80	21.48 ± 0.87			
Fat mass	Pre	15.36 ± 4.58	16.12 ± 4.29	0.197	5.795 *	0.356
(kg)	Post	14.75 ± 4.67	15.75 ± 4.34			
Body mass index	Pre	19.27 ± 1.85	19.67 ± 1.49	0.271	0.717	0.034
(kg/m^2^)	Post	19.22 ± 1.84	19.60 ± 1.46			
Percent fat	Pre	27.81 ± 6.14	28.64 ± 5.22	0.155	4.212	0.239
(%)	Post	26.84 ± 6.47	28.04 ± 5.38			
Waist/hip ratio	Pre	0.80 ± 0.03	0.79 ± 0.02	0.009	19.997 **	6.703 *
	Post	0.79 ± 0.03	0.79 ± 0.32			

All data represents the mean ± standard deviation. 3DEG and CON mean 3D-platform exercised group, and control group. * and ** represent *p* < 0.05 and *p* < 0.01, respectively.

**Table 4 medicina-56-00351-t004:** Comparative results of TMG variables from rectus abdominis and erector spinae.

Items (Units)		Time (T)	Group (G)		ANOVA (*F*)		
			3DEG	CON	G	T	G × T
Rectus	left Tc	Pre	23.73 ± 11.06	26.22 ± 7.80	0.007	0.741	0.592
abdominis	(ms)	Post	29.61 ± 13.81	26.55 ± 10.67			
muscle	left Dm	Pre	0.92 ± 0.76	1.97 ± 1.61	3.130	3.004	0.294
	(mm)	Post	1.86 ± 1.55	2.47 ± 1.66			
	right Tc	Pre	26.86 ± 6.55	28.97 ± 12.61	0.066	0.263	1.150
	(ms)	Post	28.40 ± 9.98	24.61 ± 9.61			
	right Dm	Pre	1.13 ± 1.42	2.04 ± 1.37	3.573	0.684	0.019
	(mm)	Post	1.38 ± 1.28	2.39 ± 1.59			
Erector	left Tc	Pre	20.16 ± 18.19	25.67 ± 16.39	0.834	0.100	0.126
spinae	(ms)	Post	20.00 ± 9.32	28.49 ± 31.95			
muscle	left Dm	Pre	0.60 ± 0.51	0.99 ± 0.79	0.129	0.140	2.149
	(mm)	Post	0.97 ± 0.96	0.77 ± 0.71			
	right Tc	Pre	13.17 ± 4.23	18.48 ± 11.43	1.279	7.421 *	0.177
	(ms)	Post	24.55 ± 15.50	34.01 ± 33.92			
	right Dm	Pre	0.54 ± 0.37	0.78 ± 0.85	0.016	0.378	1.581
	(mm)	Post	0.84 ± 0.84	0.68 ± 0.68			

All data represents mean ± standard deviation. * represents *p* < 0.05. Here, TMG, Tc and Dm mean tensiomyography, contraction time and maximum displacement, respectively.

**Table 5 medicina-56-00351-t005:** Comparative results of isokinetic moments from trunk flexor and trunk extensor.

Items (Units)		Time (T)	Group (G)		ANOVA (*F*)		
			3DEG	CON	G	T	G × T
Flexor	Pt	Pre	129.33 ± 16.97	127.33 ± 16.87	0.607	3.757	0.468
	(Nm)	Post	137.41 ± 15.27	130.55 ± 12.74			
Extensor	Pt	Pre	118.25 ± 28.81	132.00 ± 32.80	0.523	0.001	0.962
	(Nm)	Post	122.58 ± 30.04	127.88 ± 35.13			
Flexor	Wr	Pre	109.16 ± 40.68	116.66 ± 36.47	0.602	3.820	0.381
	(Nm)	Post	116.33 ± 23.91	130.44 ± 32.34			
Extensor	Wr	Pre	76.33 ± 34.96	89.11 ± 29.24	0.132	4.551 *	2.283
	(Nm)	Post	93.50 ± 34.66	91.00 ± 36.49			

All data represents mean ± standard deviation. * represents *p* < 0.05. Pt and Wr mean peak torque and work per repetition, respectively.

**Table 6 medicina-56-00351-t006:** Comparative results of trunk flexibilities.

Items (Units)	Time (T)	Group (G)		ANOVA (*F*)		
		3DEG	CON	G	T	G × T
Forward flexibility	Pre	12.31 ± 4.66	15.61 ± 4.04	2.161	0.324	1.059
(cm)	Post	12.73 ± 5.30	15.49 ± 4.53			
Backward flexibility	Pre	44.92 ± 7.08	47.22 ± 7.95	0.096	5.204 *	1.341
(cm)	Post	49.00 ± 7.03	48.56 ± 7.45			

All data represents mean ± standard deviation. * represents *p* < 0.05.

**Table 7 medicina-56-00351-t007:** Comparative results of back pain degrees.

Items	Time (T)	Group (G)		ANOVA (*F*)		
		3DEG	CON	G	T	G × T
Back pain	Pre	5.55 ± 2.41	5.93 ± 1.91	1.435	25.444 **	0.001
	Post	1.08 ± 1.11	5.28 ± 3.64			
Night pain	Pre	3.20 ± 1.89	1.40 ± 1.57	5.194 *	6.631 *	0.869
	Post	1.66 ± 2.07	0.68 ± 0.75			
Exercise	Pre	3.63 ± 3.16	4.14 ± 3.05	0.169	12.249 **	0.023
	Post	1.38 ± 2.16	1.69 ± 2.58			
Stiffness	Pre	2.32 ± 2.26	1.24 ± 1.49	1.795	3.534	0.379
	Post	1.48 ± 1.41	0.82 ± 0.98			
Walking freedom	Pre	2.05 ± 2.07	1.40 ± 1.20	0.609	3.297	0.200
	Post	1.43 ± 1.77	1.02 ± 1.15			
Walking discomfort	Pre	2.20 ± 2.17	1.51 ± 1.41	0.680	5.172 *	0.078
	Post	1.48 ± 1.89	0.94 ± 1.41			
Standing still	Pre	2.53 ± 2.43	1.66 ± 2.33	0.886	2.643	0.077
	Post	1.60 ± 2.08	1.00 ± 1.05			
Twisting pain	Pre	2.99 ± 2.40	1.92 ± 1.71	0.266	2.134	2.772
	Post	1.80 ± 1.98	2.00 ± 2.17			
Hard chair	Pre	3.63 ± 2.69	3.59 ± 2.84	0.152	7.124 *	0.858
	Post	1.98 ± 1.98	2.79 ± 2.26			
Soft chair	Pre	2.63 ± 2.26	1.31 ± 1.35	2.244	5.732 *	0.498
	Post	1.70 ± 1.78	0.81 ± 0.91			
Lying down	Pre	3.96 ± 3.20	4.50 ± 3.85	0.162	6.950 *	0.001
	Post	2.30 ± 2.67	2.81 ± 3.56			

All data represents mean ± standard deviation. * and ** represent *p* < 0.05 and *p* < 0.01, respectively.

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
