# Peer review of "Effects of 3D Moving Platform Exercise on Physiological Parameters and Pain in Patients with Chronic Low Back Pain"

_medicina, 2020, doi:10.3390/medicina56070351_

Round 1
Reviewer 1 Report
The study conducted by Kim et al. tested a hypothesis that exercise intervention using a 3D moving platform would be an effective option for improving chronic low back pain in young females. The authors argued that the 3D moving exercise was superior to the traditional physical therapeutics in improving flexibility, strength, and body composition, which are favorable in reducing the magnitude of low back pain. Overall, the study design and methods are sound. The manuscript has a logical flow and was generally well written, yet perhaps a professional English editing service would improve clarity for better communication. Major and minor comments are listed below.
<Major comments>
- The study design includes two main factors (intervention X time), so a two-way ANOVA should be performed. Also, in tables, displaying p-values would be better than showing Z-scores. Z-score can move to the text.
- The introductory statement of the 3D moving platform (Line 57-58) should be described better because it is a machine that has a 3D element to engage in a sort of core exercise regimen or program. Considering the dynamic nature of such a motion training device, patients would need to perform it safely in the presence of a therapist.
- The sentence, in line 63-66, should be re-write. Is this study really interested in using a core stability exercise program to measure pain, trunk flexibility, static/dynamic muscle contractions, or is this to study the efficacy of such a program using a new platform?
- (Line 80-81) It should be clarified what questionnaires were used. Were validated questionnaires about chronic low back pain used or questionnaires designed by the authors?
- What does the Visual Analogue Scale questionnaire measure? Assuming it should specify it's a back pain intensity scale, a reference should be inserted.
- Where was the sensor tip placed in this study? Was the same location for erector spinae and rectus abdominus used for all participants? How was location determined?
- Figure 2 is hard to ready because of the low-quality image. These appear to be large differences in the right and left side and should be accounted for as potential strength imbalances or muscle damage as claimed that have been shown to potentially cause pain. This change in the graphs should be addressed better in the text for why the change in shape as well as the duration is important as an outcome measure and what it means for clinical application. The representative graph should be replaced by a better version if authors would like to keep it in the manuscript.
- For the dynamic muscle contraction measurements, were other bony landmarks used or was the distance above and below the patella measured to ensure same placement for all participants, i.e. tibial pad placed on tibial tuberosity or 1 cm below tibial tuberosity? Or were scapulae measured to ensure pad was central on each participants' scapula?
- The rehab program performed by control subjects needs to be explained. It says that "the rehabilitation program consisted of various types of exercise for softening the body and for strengthening the paraspinal muscle." What are these exercises? Are they listed in a table or chart elsewhere? How long the control group performed such an exercise program?
- The results have intra-contradiction in body composition measures. The intervention group had a significant decrease in WHR that is a measure of body composition. BIA can show varying results based on hydration status. Did participants have to avoid diuretic drinks for 7 days prior to the post-intervention measurement or just or the baseline?
- In table 2, it would be much clear to use anatomical terms and actual movements as opposed to cues - For example, "drawing in" I assume is the activation of the rectus adbominus to anteriorly tilt the pelvis to create a neutral spine position which reduces the natural arch.
- (Lines 257-258) How did the 3D platform exercise reduce pain during activity rather than rest if LBP was significantly decreased while during the night (night pain) and while lying down after the intervention?
- (Lines 285-287) These differences between right and left ES and RA show strength imbalances and have been reported to result in pain. Were the strength imbalances observed in the intervention group the same as those in the control group or greater or less imbalanced? Could this account for the observed decreases in perceived pain? These topics should be further discussed.
- It was discussed that the 3D moving platform exercise was more effective for extensor than flexor muscles. The status of the malalignment of the study participants should be discussed. Did these participants already possess a lordotic or kyphotic spine? If so those with lordosis are typically better able to extend than flex the trunk and those with kyphosis can flex the trunk pretty well. Perhaps, training those motions would show greater improvements for motion already more familiar/easy to do for participants.
- (Lines 298-301) The statement should be linked to specific results to show this interpretation and what the outcome measures mean in terms of application and clinical relevance.
- (Lines 305-306) It is unclear how did this platform provides aerobic conditioning effects from the exercises that were described in the table earlier (Tab 2).
<Minor comments>
Line 33. Almost people -> Most individuals
Line 35. There are many causes of back pain, but it is likely to occur due to the absence of physical activity, which causes weakness of the... -> Among many factors causing low back pain, a lack of physical activity has been viewed as a primary risk factor, which results in weakness of the...
Line 43. unhealthy lifestyle habits -> unhealthy lifestyle
Line 59. "more effective" than what?
Author Response
Answers to reviewer’s comments
Thank you for your kind advice and comments for publication in Medicina. We revised our manuscript as per your comments. We represented the specific modifications in response to the comments by blue-letters in our manuscript. We sincerely appreciate your comments because your comments make our manuscript better.
Reviewer 1:
<Major comments>
#1. Comments and Suggestions:
The study design includes two main factors (intervention × time), so a two-way ANOVA should be performed. Also, in tables, displaying p-values would be better than showing Z-scores. Z-score can move to the text.
#1. Response: Thank you for what the reviewer has pointed out above comments. According to your suggestion, all statistical methods were changed to two-way ANOVA, and when looking at the p value of interaction in the results. Details of these revised results are as follows: on Line 214 to 260 of the original manuscript.
“ 3. Results
3.1. Difference in anthropometric indices
There were no differences in baseline characteristics between 3DEG and CON, which indicates homogeneity was established. As shown in Table 3, the participants showed similar results, except for fat mass and waist/hip ratio (WHR). In detail, although the Δ% of fat mass in CON increased ≈ 2.3%, that of 3DEG decreased ≈ 4% (not shown in the table). This indicates that the platform exercise provided a greater amount of exercise that can further reduce fat. These results indicate that there was a significant difference in time (P < 0.05). Similar to the results of fat mass, the WHR of CON did not change significantly after 8 weeks, whereas that of 3DEG decreased ≈ 1.25% (not shown in the table), which was significantly different in time (P < 0.01) and group by time (P < 0.05).
Table 3. Comparative results of body composition.
|
Items (units) |
Time (T) |
Group (G) |
ANOVA (F) |
||||
|
3DEG |
CON |
G |
T |
G×T |
|||
|
Weight |
Pre |
54.41 ± 5.85 |
55.67 ± 4.93 |
0.291 |
1.208 |
0.008 |
|
|
(kg) |
Post |
54.13 ± 5.76 |
55.43 ± 4.66 |
||||
|
Muscle mass |
Pre |
21.07 ± 1.91 |
21.32 ± 1.23 |
0.092 |
1.799 |
0.072 |
|
|
(kg) |
Post |
21.32 ± 1.80 |
21.48 ± 0.87 |
||||
|
Fat mass |
Pre |
15.36 ± 4.58 |
16.12 ± 4.29 |
0.197 |
5.795* |
0.356 |
|
|
(kg) |
Post |
14.75 ± 4.67 |
15.75 ± 4.34 |
||||
|
Body mass index |
Pre |
19.27 ± 1.85 |
19.67 ± 1.49 |
0.271 |
0.717 |
0.034 |
|
|
(kg/m2) |
Post |
19.22 ± 1.84 |
19.60 ± 1.46 |
||||
|
Percent fat |
Pre |
27.81 ± 6.14 |
28.64 ± 5.22 |
0.155 |
4.212 |
0.239 |
|
|
(%) |
Post |
26.84 ± 6.47 |
28.04 ± 5.38 |
||||
|
Waist/hip ratio |
Pre |
0.80 ± 0.03 |
0.79 ± 0.02 |
0.009 |
19.997** |
6.703* |
|
|
Post |
0.79 ± 0.03 |
0.79 ± 0.32 |
|||||
|
All data represents the mean ± standard deviation. 3DEG and CON mean 3D-platform exercised group, and control group. * and ** represent P < 0.05 and P < 0.01, respectively. |
|||||||
3.2. Effect of 3D platform exercise on static muscle contraction
As shown in Table 4, there were no significant changes in the left or right Tc and Dm of the CON. Similarly, there were no significant changes in the left or right Tc and Dm of the 3DEG after 8 weeks. However, the increased right Tc of the erector spinae in CON was greater than that of the erector spinae in 3DEG. These results indicated that there was a significant difference in time (P < 0.05). This indicates that the platform exercise provided a greater amount of exercise that can increase the contraction of muscles and improve balanced development of the left (20.00 ± 9.32 ms) and right (24.5 ± 15.55 ms) erector spinae muscles after 8 weeks.
Table 4. Comparative results of TMG variables from rectus abdominis and erector spinae.
|
Items (units) |
Time (T) |
Group (G) |
ANOVA (F) |
|||||
|
3DEG |
CON |
G |
T |
G×T |
||||
|
Rectus |
left Tc |
Pre |
23.73 ± 11.06 |
26.22 ± 7.80 |
0.007 |
0.741 |
0.592 |
|
|
abdominis |
(ms) |
Post |
29.61 ± 13.81 |
26.55 ± 10.67 |
||||
|
muscle |
left Dm |
Pre |
0.92 ± 0.76 |
1.97 ± 1.61 |
3.130 |
3.004 |
0.294 |
|
|
(mm) |
Post |
1.86 ± 1.55 |
2.47 ± 1.66 |
|||||
|
right Tc |
Pre |
26.86 ± 6.55 |
28.97 ± 12.61 |
0.066 |
0.263 |
1.150 |
||
|
(ms) |
Post |
28.40 ± 9.98 |
24.61 ± 9.61 |
|||||
|
Right Dm |
Pre |
1.13 ± 1.42 |
2.04 ± 1.37 |
3.573 |
0.684 |
0.019 |
||
|
(mm) |
Post |
1.38 ± 1.28 |
2.39 ± 1.59 |
|||||
|
Erector |
left Tc |
Pre |
20.16 ± 18.19 |
25.67 ± 16.39 |
0.834 |
0.100 |
0.126 |
|
|
spinae |
(ms) |
Post |
20.00 ± 9.32 |
28.49 ± 31.95 |
||||
|
muscle |
left Dm |
Pre |
0.60 ± 0.51 |
0.99 ± 0.79 |
0.129 |
0.140 |
2.149 |
|
|
(mm) |
Post |
0.97 ± 0.96 |
0.77 ± 0.71 |
|||||
|
right Tc |
Pre |
13.17 ± 4.23 |
18.48 ± 11.43 |
1.279 |
7.421* |
0.177 |
||
|
(ms) |
Post |
24.55 ± 15.50 |
34.01 ± 33.92 |
|||||
|
right Dm |
Pre |
0.54 ± 0.37 |
0.78 ± 0.85 |
0.016 |
0.378 |
1.581 |
||
|
(mm) |
Post |
0.84 ± 0.84 |
0.68 ± 0.68 |
|||||
|
All data represents mean ± standard deviation. * and ** represent P < 0.05 and P < 0.01, respectively. Here, TMG, Tc and Dm mean tensiomyography, contraction time and maximum displacement, respectively. |
||||||||
3.3. Effect of 3D platform exercise on dynamic muscle contraction
As shown in Table 5, there were no significant changes in most variables of isokinetic moments at 30°/s in both groups after 8 weeks. However, the Wr of trunk extensor at 90°/s in the 3DEG was significantly increased in the trunk extensor, but this was not changed in the CON. More specifically, although the Δ% of the Wr of trunk extensor in CON increased ≈ 2.1%, that of 3DEG increased ≈ 22.5% (not shown in the table). This indicates that the platform exercise provided a greater amount of exercise that can increase dynamic contractions of the trunk extensor. These results show that there was a significant difference in time (P < 0.05).
Table 5. Comparative results of isokinetic moments from trunk flexor and trunk extensor.
|
Items (units) |
Time (T) |
Group (G) |
ANOVA (F) |
|||||
|
3DEG |
CON |
G |
T |
G×T |
||||
|
Flexor |
Pt |
Pre |
129.33 ± 16.97 |
127.33 ± 16.87 |
0.607 |
3.757 |
0.468 |
|
|
(Nm) |
Post |
137.41 ± 15.27 |
130.55 ± 12.74 |
|||||
|
Extensor |
Pt |
Pre |
118.25 ± 28.81 |
132.00 ± 32.80 |
0.523 |
0.001 |
0.962 |
|
|
(Nm) |
Post |
122.58 ± 30.04 |
127.88 ± 35.13 |
|||||
|
Flexor |
Wr |
Pre |
109.16 ± 40.68 |
116.66 ± 36.47 |
0.602 |
3.820 |
0.381 |
|
|
(Nm) |
Post |
116.33 ± 23.91 |
130.44 ± 32.34 |
|||||
|
Extensor |
Wr |
Pre |
76.33 ± 34.96 |
89.11 ± 29.24 |
0.132 |
4.551* |
2.283 |
|
|
(Nm) |
Post |
93.50 ± 34.66 |
91.00 ± 36.49 |
|||||
|
All data represents mean ± standard deviation. * and ** represent P < 0.05 and P < 0.01, respectively. Pt and Wr mean peak torque and work per repetition, respectively. |
||||||||
3.4. Effect of 3D platform exercise on trunk flexibilities
As shown in Table 6, although the trunk forward flexibility of 3DEG tended to increase, it tended to decrease in the CON, although there was not a significant difference. However, the trunk backward flexibility in both groups tended to increase. In detail, the Δ% of the backward flexibility in CON increased ≈ 2.8%, while that of 3DEG increased ≈ 9.1% (not shown in the table). This indicates that the platform exercise provided a greater amount of exercise that can further soften the trunk extensor muscles. These results show that there was a significant difference in time (P < 0.05).
Table 6. Comparative results of trunk flexibilities.
|
Items (units) |
Time (T) |
Group (G) |
ANOVA (F) |
||||
|
3DEG |
CON |
G |
T |
G×T |
|||
|
Forward flexibility |
Pre |
12.31 ± 4.66 |
15.61 ± 4.04 |
2.161 |
0.324 |
1.059 |
|
|
(cm) |
Post |
12.73 ± 5.30 |
15.49 ± 4.53 |
||||
|
Backward flexibility |
Pre |
44.92 ± 7.08 |
47.22 ± 7.95 |
0.096 |
5.204* |
1.341 |
|
|
(cm) |
Post |
49.00 ± 7.03 |
48.56 ± 7.45 |
||||
|
All data represents mean ± standard deviation. * and ** represent P < 0.05 and P < 0.01, respectively. |
|||||||
3.5. Effect of 3D platform exercise on visual analogue scale
As shown in Table 7, the VAS scores in both groups tended to improve, although the VAS scores showed a significant change in some variables, while others did not. In detail, the Δ% of back pain in CON decreased ≈ 53.8%, while that of 3DEG decreased ≈ 80.5% (not shown in the table). The Δ% in feeling pain at night, during exercise, walking, sitting in a hard chair, sitting in a soft chair, and lying down in CON were changed by ≈ 9.2%, ≈ -31.6%, ≈ -13.1%, ≈ -26.4%, ≈ -21.2% and ≈ -16.8%, respectively. The Δ% in feeling pain at night, during exercise, walking, sitting in a hard chair, sitting in a soft chair, and lying down in 3DEG were changed by ≈ -48.2%, ≈ -62.1%, ≈ -32.9%, ≈ -45.4%, ≈ -35.3% and ≈ -41.9%, respectively (not shown in the table). This indicates that the platform exercise provided a greater reduction of pain for activities that are done on a daily basis. There were significant differences in back pain (P < 0.05, within time), night pain (P < 0.05, between group; P < 0.05, within time), exercise (P < 0.01, within time), walking discomfort (P < 0.05, within time), hard chair (P < 0.05, within time), soft chair (P < 0.05, within time), and lying down (P < 0.05, within time), respectively.
Table 7. Comparative results of back pain degrees.
|
Items |
Time (T) |
Group (G) |
ANOVA (F) |
||||
|
3DEG |
CON |
G |
T |
G×T |
|||
|
Back pain |
Pre |
5.55 ± 2.41 |
5.93 ± 1.91 |
1.435 |
25.444** |
0.001 |
|
|
Post |
1.08 ± 1.11 |
5.28 ± 3.64 |
|||||
|
Night pain |
Pre |
3.20 ± 1.89 |
1.40 ± 1.57 |
5.194* |
6.631* |
0.869 |
|
|
Post |
1.66 ± 2.07 |
0.68 ± 0.75 |
|||||
|
Exercise |
Pre |
3.63 ± 3.16 |
4.14 ± 3.05 |
0.169 |
12.249** |
0.023 |
|
|
Post |
1.38 ± 2.16 |
1.69 ± 2.58 |
|||||
|
Stiffness |
Pre |
2.32 ± 2.26 |
1.24 ± 1.49 |
1.795 |
3.534 |
0.379 |
|
|
Post |
1.48 ± 1.41 |
0.82 ± 0.98 |
|||||
|
Walking freedom |
Pre |
2.05 ± 2.07 |
1.40 ± 1.20 |
0.609 |
3.297 |
0.200 |
|
|
Post |
1.43 ± 1.77 |
1.02 ± 1.15 |
|||||
|
Walking discomfort |
Pre |
2.20 ± 2.17 |
1.51 ± 1.41 |
0.680 |
5.172* |
0.078 |
|
|
Post |
1.48 ± 1.89 |
0.94 ± 1.41 |
|||||
|
Standing still |
Pre |
2.53 ± 2.43 |
1.66 ± 2.33 |
0.886 |
2.643 |
0.077 |
|
|
Post |
1.60 ± 2.08 |
1.00 ± 1.05 |
|||||
|
Twisting pain |
Pre |
2.99 ± 2.40 |
1.92 ± 1.71 |
0.266 |
2.134 |
2.772 |
|
|
Post |
1.80 ± 1.98 |
2.00 ± 2.17 |
|||||
|
Hard chair |
Pre |
3.63 ± 2.69 |
3.59 ± 2.84 |
0.152 |
7.124* |
0.858 |
|
|
Post |
1.98 ± 1.98 |
2.79 ± 2.26 |
|||||
|
Soft chair |
Pre |
2.63 ± 2.26 |
1.31 ± 1.35 |
2.244 |
5.732* |
0.498 |
|
|
Post |
1.70 ± 1.78 |
0.81 ± 0.91 |
|||||
|
Lying down |
Pre |
3.96 ± 3.20 |
4.50 ± 3.85 |
0.162 |
6.950* |
0.001 |
|
|
Post |
2.30 ± 2.67 |
2.81 ± 3.56 |
|||||
|
All data represents mean ± standard deviation. * and ** represent P < 0.05 and P < 0.01, respectively. |
|||||||
”
#2. Comments and Suggestions:
The introductory statement of the 3D moving platform (Line 57-58) should be described better because it is a machine that has a 3D element to engage in a sort of core exercise regimen or program. Considering the dynamic nature of such a motion training device, patients would need to perform it safely in the presence of a therapist.
#2. Response: According to your suggestion, we inserted the sentences as follows: “The 360-degree rotational motion function of the 3D moving platform is designed to fit the natural spiral motion of the body and strengthen the muscles around the body by transmitting the exercise power to the deep muscles that the existing linear reciprocating motion does not reach. This ensures stability by establishing various three-dimensional railings, and can maximize pain relief as well as corrective treatment through muscle extension and contraction. It is a device that helps to properly align the deformed joint and the twisted spine by applying a sling or harness assistive device.” on Line 57 to 58 of the original manuscript.
#3. Comments and Suggestions:
The sentence, in line 63-66, should be re-write. Is this study really interested in using a core stability exercise program to measure pain, trunk flexibility, static/dynamic muscle contractions, or is this to study the efficacy of such a program using a new platform?
#3. Response: Thank you for what the reviewer has pointed out above comments. According to your suggestion, we inserted the sentence as follows: “As such, CSE can improve various properties related to the lumbar joint, and furthermore, it is thought that the 3D platform developed for CSE can also perform the functions of CSE. However, until now, it is not known whether the use of this device has a similar effect to CSE.” On Line 241 to 242 of the new changed manuscript.
#4. Comments and Suggestions:
(Line 80-81) It should be clarified what questionnaires were used. Were validated questionnaires about chronic low back pain used or questionnaires designed by the authors?
#4. Response: Thank you for what the reviewer has pointed out above comments. According to your suggestion, we corrected the sentence as follows: “Prior to the study, the participants received detailed explanations regarding all of the procedures in this study and were then asked to complete questionnaires, which included basic demographic questions and a visual analogue scale (VAS) for CLBP.” On Line 80 to 81 of the original manuscript.
#5. Comments and Suggestions:
What does the Visual Analogue Scale questionnaire measure? Assuming it should specify it's a back pain intensity scale, a reference should be inserted.
#5. Response: Thank you for what the reviewer has pointed out above comments. According to your suggestion, we corrected and inserted the sentence as follows: “All of the participants signed an informed consent form and completed a self-report questionnaire including a VAS, which is a tool for measuring the degree of pain felt in the lower back from a comfortable position to an active position [22].” on Line 99 to 101 of the original manuscript.
Reference: 22. Ogon, M.; Krismer, M.; Söllner, W.; Kantner-Rumplmair, W.; Lampe A. Chronic low back pain measurement with visual analogue scales in different settings. Pain 1996, 64, 425-428. doi: 10.1016/0304-3959(95)00208-1.
#6. Comments and Suggestions:
Where was the sensor tip placed in this study? Was the same location for erector spinae and rectus abdominus used for all participants? How was location determined?
#6. Response: Thank you for what the reviewer has pointed out above comments. This sentence would be likely to confuse the reader, so we inserted new sentences as follows: “TMG assesses muscle mechanical responses based on radial muscle belly displacement induced by a single electrical stimulus between the proximal and distal parts of the rectus abdominis and erector spinae. In the case of the rectus abdominis, the sensor tip was placed at a point 3 cm away from the left and right sides of the navel. Electrodes were attached 3 cm apart, proximal and distal from the sensor tip, which served as the center point. The degree of muscle contraction was measured at the radial muscle belly. The position of the sensor tip of the erector spinae was determined to be 5 cm above the lumbosacral joint. It was positioned 3 cm away from the left and right sides of that point, and the electrodes were attached 3 cm apart, proximal and distal from the sensor tip. The electric stimulation provided under increasing electrical current intensities was between 10–65 mA and the length of the stimulation was one millisecond (ms). An isometric contraction was generated by the electrical stimulation. Electric stimulation was given in 10mA increments until maximal displacement was reached.” On Line 131 of the original manuscript.
In general, the location on the manual for measuring the muscle belly of erector spinae and rectus abdominus using TMG is shown in the figure below.
#7. Comments and Suggestions:
Figure 2 is hard to ready because of the low-quality image. These appear to be large differences in the right and left side and should be accounted for as potential strength imbalances or muscle damage as claimed that have been shown to potentially cause pain. This change in the graphs should be addressed better in the text for why the change in shape as well as the duration is important as an outcome measure and what it means for clinical application. The representative graph should be replaced by a better version if authors would like to keep it in the manuscript.
#7. Response: Thank you for what the reviewer has pointed out above comments. According to your comments, we changed the image diagram in Fig. 2 and attached again.
#8. Comments and Suggestions:
For the dynamic muscle contraction measurements, were other bony landmarks used or was the distance above and below the patella measured to ensure same placement for all participants, i.e. tibial pad placed on tibial tuberosity or 1 cm below tibial tuberosity? Or were scapulae measured to ensure pad was central on each participants' scapula?
#8. Response: Thank you for your correction request. According to your comments, we inserted new exploration as follows: “....... For the TEF test, the upper leg and the lower leg should be fixed to prevent anterior protrusion of the lower limb. The fixed position of the upper leg was such that the bottom of the upper leg pad was aligned with the top of the patella, and in the case of lower leg, the top of the lower leg pad was aligned with the bottom of the patella. Once those pads were aligned, the locking lever was secured. In order to fix the upper body, a pad was wrapped around the chest while the subject held the handles for additional stability. At this time, the lower surface of the upper body pad was measured and then fixed to coincide with the inferior angle of the scapula.” On Line 154 to 159 of the original manuscript.
#9. Comments and Suggestions:
The rehab program performed by control subjects needs to be explained. It says that "the rehabilitation program consisted of various types of exercise for softening the body and for strengthening the paraspinal muscle." What are these exercises? Are they listed in a table or chart elsewhere? How long the control group performed such an exercise program?
#9. Response: Thank you for what the reviewer has pointed out above comments. According to your suggestion, we inserted the sentences as follows: “They participated in the rehabilitation program while the platform was operating, while CON participated in the rehabilitation program when the platform was not operating. The rehabilitation program on the 3D platform consisted of various types of exercises for stretching the core muscles and for strengthening the paraspinal muscles.” This is presented in Table 2, which is described in detail.” On Line 179 to 182 of the original manuscript.
#10. Comments and Suggestions:
The results have intra-contradiction in body composition measures. The intervention group had a significant decrease in WHR that is a measure of body composition. BIA can show varying results based on hydration status. Did participants have to avoid diuretic drinks for 7 days prior to the post-intervention measurement or just or the baseline?
#10. Response: Thank you for what the reviewer has pointed out above comments. We had written the sentences on line 108 to 109 as follows: “The participants abstained from food, exercise, and diuretic drinks for 4 hours, 12 hours, and 7 days, respectively, prior to assessments. The participants were also asked to void 30 min prior to the assessment [15].”
#11. Comments and Suggestions:
In table 2, it would be much clear to use anatomical terms and actual movements as opposed to cues - For example, "drawing in" I assume is the activation of the rectus adbominus to anteriorly tilt the pelvis to create a neutral spine position which reduces the natural arch.
#11. Response: Thank you for what the reviewer has pointed out above comments. According to your suggestion, In Table 2, incorrect anatomical terms were corrected, and motion terms were also corrected.
#12. Comments and Suggestions:
(Lines 257-258) How did the 3D platform exercise reduce pain during activity rather than rest if LBP was significantly decreased while during the night (night pain) and while lying down after the intervention?
#12. Response: Thank you for what the reviewer has pointed out above comments. According to your suggestion, we inserted the sentences in the discussion part as follows: "It is thought that the pain level felt through the expansion of the narrow nerve path was reduced because the CSE performed on the 3D platform helped the spine alignment by applying more muscle stretching and further stimulating the deep muscles.” on Line 326 page.
#13. Comments and Suggestions:
(Lines 285-287) These differences between right and left ES and RA show strength imbalances and have been reported to result in pain. Were the strength imbalances observed in the intervention group the same as those in the control group or greater or less imbalanced? Could this account for the observed decreases in perceived pain? These topics should be further discussed.
#13. Response: Thank you for what the reviewer has pointed out above comments. According to your suggestion, we inserted the sentences as follows: "In other words, by exercising on the 3D platform, we could see the balanced effect that the Tc on the right rises and resembles the Tc on the left in erector spinae muscles.". This is also described in the results section as follows. “This means that the platform exercise provided the greater amount of exercise that could contract the muscles and the balanced development of the left (20.00 ± 9.32 ms) and right (24.5 ± 15.55 ms) erector spinae muscles after 8 weeks.” around Line 231 page in the original manuscript.
#14. Comments and Suggestions:
It was discussed that the 3D moving platform exercise was more effective for extensor than flexor muscles. The status of the malalignment of the study participants should be discussed. Did these participants already possess a lordotic or kyphotic spine? If so those with lordosis are typically better able to extend than flex the trunk and those with kyphosis can flex the trunk pretty well. Perhaps, training those motions would show greater improvements for motion already more familiar/easy to do for participants.
#14. Response: Thank you for what the reviewer has pointed out above comments. According to your suggestion, we inserted the sentences as follows: “The subjects in this study had no structural deformations such as a lordotic or kyphotic spine. However, when observing that the left and right paraspinal muscles become balanced through the 3D platform movement, it is possible that it can provide help to some extent for individuals with structural diseases. Ultimately, it is thought that the static muscle contraction ability and the dynamic muscle contraction ability developed simultaneously through the core exercises on the 3D platform.” On Line 351 in the original manuscript.
#15. Comments and Suggestions:
(Lines 298-301) The statement should be linked to specific results to show this interpretation and what the outcome measures mean in terms of application and clinical relevance.
#15. Response: Thank you for what the reviewer has pointed out above comments. According to your suggestion, we inserted the sentences as follows: " In other words, it was observed that the muscles in the back of the lumbar region were improved, whether static or dynamic, and characteristically, the part lacking muscle function on either side of the back was improved. This provides information that can be used clinically when applying the 3D platform to patients with low back pain.” On Line 352 in the original manuscript.
#16. Comments and Suggestions:
(Lines 305-306) It is unclear how did this platform provides aerobic conditioning effects from the exercises that were described in the table earlier (Tab 2).
#16. Response: Thank you for what the reviewer has pointed out above comments. According to your suggestion, we inserted the sentences as follows: " It is thought that 3D platform exercise provides deeper muscle activation on the rotating platform, and provides a higher metabolic energy consumption effect than CON, which was in motion while the platform was stationary.” On the Line 305 in the original manuscript.
<Minor comments>
#1. Comments and Suggestions:
Line 33. Almost people -> Most individuals
#1. Response: Thank you for what the reviewer has pointed out above comments. According to your suggestion, we changed the phrase.
#2. Comments and Suggestions:
Line 35. There are many causes of back pain, but it is likely to occur due to the absence of physical activity, which causes weakness of the... -> Among many factors causing low back pain, a lack of physical activity has been viewed as a primary risk factor, which results in weakness of the...
#2. Response: Thank you for what the reviewer has pointed out above comments. According to your suggestion, we changed the sentence.
#3. Comments and Suggestions:
Line 43. unhealthy lifestyle habits -> unhealthy lifestyle
#3. Response: Thank you for what the reviewer has pointed out above comments. According to your suggestion, we changed the phrase.
#4. Comments and Suggestions:
Line 59. "more effective" than what?
#4. Response: Thank you for what the reviewer has pointed out above comments. According to your suggestion, we erased ‘more’ to minimize ambiguous expressions.
We’ve got the English Editing Service through https://www.mdpi.com/authors/english.
Thank you for your comments, we represented the modifications in response to your comments.
July 12, 2020
Reviewer 2 Report
This is an interesting paper, from the exercise type point of view. However, the mechanisms of this exercise approach are rather explored in this paper,with a multitude of outcome measures, rather than known.
In the title, the phrase ‘physical conditions’ requires changing. Consider amending to the phrase ‘physiological parameters’.
Line 33: Almost all people… (underlined word is missing)
Line 56: Insert reference after the word ‘’platform”
Line 71: Can the authors comment on the fact that the patients included were particularly young. Had they all developed chronic LBP at such a young age?
Figure 1: Follow up left box: The phrase “move to a strange place” requires removing
Lines 263-264: change to ‘although it remained unchanged’
Lines 263 / 264-266 / 284-5: changes were not of the same amount in all the measures. Perhaps the authors need to present greater changes first.
Lines 277-80: It is difficult to understand why the 3D moving platform exercise is thought to be working in a similar fashion to core stability exercise, preferentially activating the core trunk muscles in relation to the more superficial ones. This is a serious issue that requires rephrasing in several parts of the paper or some adequate explanation of the link between the 2 forms of exercise.
Limitations:
-No power analysis performed and relatively small sample used.
-Results apply only to females.
-No long term follow up.
Author Response
Answers to reviewer’s comments
Thank you for your kind advice and comments for publication in Medicina. We re-revised the manuscript as per your comments. We represented the specific modifications in response to the comments by blue-letters in our manuscript. We sincerely appreciate your comments because your comments make our manuscript better.
Reviewer 2:
#1. Comments and Suggestions:
In the title, the phrase ‘physical conditions’ requires changing. Consider amending to the phrase ‘physiological parameters’.
#1. Response: Thank you for what the reviewer has pointed out above comments. According to your suggestion, we changed the title from ‘physical conditions’ to ‘physiological parameters’.
#2. Comments and Suggestions:
Line 33: Almost all people… (underlined word is missing)
#2. Response: Thank you for what the reviewer has pointed out above comments. According to your suggestion, we changed the words.
#3. Comments and Suggestions:
Line 56: Insert reference after the word ‘’platform”
#3. Response: Thank you for what the reviewer has pointed out above comments. The 3D platform exercise we were trying to study was the first one to try, so there are no references.
#4. Comments and Suggestions:
Line 71: Can the authors comment on the fact that the patients included were particularly young. Had they all developed chronic LBP at such a young age?
#4. Response: Thank you for what the reviewer has pointed out above comments. According to your question, we can reply as follows: “Low back pain is not related to age. In the recruitment process, we were only trying to limit ourselves to women who met the conditions of the subjects. In my opinion, the participants in this study complained of low back pain because they had no exercise habits and had only sedentary lifestyle.”
#5. Comments and Suggestions:
Figure 1: Follow up left box: The phrase “move to a strange place” requires removing
#5. Response: Thank you for what the reviewer has pointed out above comments. According to your suggestion, we deleted the “move to a strange place” you pointed out.
#6. Comments and Suggestions:
Lines 263-264: change to ‘although it remained unchanged’
#6. Response: Thank you for what the reviewer has pointed out above comments. According to your suggestion, we changed the phrase on page 304 to 305.
#7. Comments and Suggestions:
Lines 263 / 264-266 / 284-5: changes were not of the same amount in all the measures. Perhaps the authors need to present greater changes first.
#7. Response: Thank you for what the reviewer has pointed out above comments. According to your suggestion, we changed the sentences as follows: " After 8 weeks of platform exercise, the fat mass and WHR in the 3DEG was improved, although it remained unchanged in CON. Furthermore, the platform exercise improved the erector spinae Tc of the static muscle contraction, extensor Wr of the dynamic muscle contraction, trunk backward flexibility, and some of VAS in the 3DEG, but did not in CON.” On the Line 262 to 266.
#8. Comments and Suggestions:
Lines 277-80: It is difficult to understand why the 3D moving platform exercise is thought to be working in a similar fashion to core stability exercise, preferentially activating the core trunk muscles in relation to the more superficial ones. This is a serious issue that requires rephrasing in several parts of the paper or some adequate explanation of the link between the 2 forms of exercise.
#8. Response: Thank you for what the reviewer has pointed out above comments. According to your suggestion, we inserted the sentences as follows: " Akuthota et al. [47] suggested that core stability is essential for proper load balance within the spine, pelvis, and kinetic chain. The so-called core is the group of trunk muscles that surround the spine and abdominal viscera.”
- Akuthota, V.; Ferreiro, A.; Moore, T.; Fredericson, M. Core stability exercise principles. Curr. Sports Med. Rep. 2008, 7, 39-44. doi: 10.1097/01.CSMR.0000308663.13278.69.
#9. Comments and Suggestions:
Limitations:
-No power analysis performed and relatively small sample used.
-Results apply only to females.
-No long term follow up.
#9. Response: Thank you for what the reviewer has pointed out above comments. According to your suggestion, we inserted the sentences as follows: " These changes observed in this study included subjects who chronically complained of low back pain due to the narrowing of the neural tube from the deformation of the muscles and ligaments around the lumbosacral joint. However, it was confirmed that the CSE exercise on the 3D platform performed for 8 weeks can effectively activate the paraspinal muscles by providing greater stimulation and RoM around the lumbosacral joint than normal CSE. However, this study has some limitations in that the number of subjects were small, consisted entirely of women, and was conducted in a rather short period of 8 weeks. Addressing the three limitations mentioned above may be helpful for future studies.”
We’ve got the English Editing Service through https://www.mdpi.com/authors/english, again.
Thank you for your comments, we represented the modifications in response to your comments.
July 12, 2020
